# Epigenomic and Proteomic Changes in Fetal Spleens Persistently Infected with Bovine Viral Diarrhea Virus: Repercussions for the Developing Immune System, Bone, Brain, and Heart

**DOI:** 10.3390/v14030506

**Published:** 2022-02-28

**Authors:** Hanah M. Georges, Hana Van Campen, Helle Bielefeldt-Ohmann, Thomas R. Hansen

**Affiliations:** 1Animal Reproduction and Biotechnology Laboratory, Department of Biomedical Sciences, Colorado State University, Fort Collins, CO 80523, USA; hanah.georges@yale.edu (H.M.G.); hana.van_campen@colostate.edu (H.V.C.); 2Australian Infections Diseases Research Centre, The University of Queensland, St. Lucia, QLD 4072, Australia; h.bielefeldtohmann1@uq.edu.au; 3School of Chemistry & Molecular Biosciences, The University of Queensland, St. Lucia, QLD 4072, Australia

**Keywords:** bovine viral diarrhea virus, fetus, immune development, osteoclastogenesis, epigenetics, methylation

## Abstract

Bovine viral diarrhea virus (BVDV) infection during early gestation results in persistently infected (PI) immunotolerant calves that are the primary reservoirs of the virus. Pathologies observed in PI cattle include congenital defects of the brain, heart, and bone as well as marked functional defects in their immune system. It was hypothesized that fetal BVDV infection alters T cell activation and signaling genes by epigenetic mechanisms. To test this, PI and control fetal splenic tissues were collected on day 245 of gestation, 170 days post maternal infection. DNA was isolated for reduced representation bisulfite sequencing, protein was isolated for proteomics, both were analyzed with appropriate bioinformatic methods. Within set parameters, 1951 hypermethylated and 691 hypomethylated DNA regions were identified in PI compared to control fetuses. Pathways associated with immune system, neural, cardiac, and bone development were associated with heavily methylated DNA. The proteomic analysis revealed 12 differentially expressed proteins in PI vs. control animals. Upregulated proteins were associated with protein processing, whereas downregulated proteins were associated with lymphocyte migration and development in PI compared to control fetal spleens. The epigenetic changes in DNA may explain the immune dysfunctions, abnormal bone formation, and brain and heart defects observed in PI animals.

## 1. Introduction

Bovine viral diarrhea virus (BVDV) is a single stranded RNA virus in the family *Flaviviridae* and Pestivirus genus, discovered in 1946 [1,2,3,4,5]. BVDV exists as two genotypes, BVDV1 and BVDV2, both having cytopathic (cp) and noncytopathic (ncp) biotypes [3,4,5,6,7]. The ncp biotype most commonly occurs in nature while cp biotypes arise from naturally occurring mutations of the ncp biotype in persistently infected (PI) animals [6,8,9,10,11]. Most BVDV infections of healthy adult cattle result in subclinical to relatively mild disease depending on the strain of virus; however, severe pathologies occur in fetal infections following the vertical transmission from dam to fetus [12,13,14]. The outcome of BVDV fetal infection is determined by the time of infection relative to the development of the fetal bovine immune system [12,13,14]. If fetal BVDV infection occurs between gestational days 42 and 125, the fetus is unable to mount a competent immune response, becomes immunotolerant and persistently infected (PI), shedding the virus throughout its postnatal life [15,16]. Fetal BVDV infections also cause a variety of congenital malformations including malformations of the brain such as cerebellar hypoplasia, hydranencephaly, and hydrocephalus, bone malformations including brachygnathism, osteopetrosis, abnormal trabecular modeling, and arthrogryposis, heart abnormalities such as septal defects, and thymic hypoplasia [12,13,17,18,19,20]. Fetuses infected after day 150 of gestation are able to mount a more mature immune response to BVDV as evidenced by the presence of virus-specific antibodies and clearance of the virus prior to birth [12].

Our group previously developed an in vivo model of BVDV PI fetal infection by inoculating pregnant BVDV naïve heifers with BVDV on day 75 of gestation [21,22]. Fetuses collected at several time points during gestation (days 82, 97, 190, and 245) revealed an approximately 2-week time period between maternal and fetal infection as determined by maternal and fetal viremia [23]. An innate interferon (IFN) response in the PI fetus occurred early in the course of the infection; however, by day 190 both the innate and adaptive branches of the fetal immune response were dramatically attenuated [24,25]. Pathological findings in the PI fetuses included bone and brain abnormalities with BVDV antigen localized in neural tissues [20,21,22,23,26,27,28,29,30,31]. It was reasoned that immune, neural, and bone pathologies were due to the inhibition of interleukins and the presence of circulating IFNs [20,26,27].

The present study aimed to determine if the inhibition of the immune system following the initial immune response and PI pathologies were due to epigenetic changes in the PI fetal DNA. It was hypothesized that spleens from PI fetuses would have increased methylation of genes and decreased abundance of proteins associated with the adaptive immune response. Spleens from PI and control fetuses collected on gestational day 245 were examined because spleens are secondary lymphoid organs which survey the blood for pathogens; the day 245 time point was chosen for its proximity to parturition (283-day gestation in the bovine) and would provide epigenetic data similar to what would be expected in postnatal animals. Reduced representation bisulfite sequencing (RRBS) and liquid chromatography and mass spectrometry (LC-MS) proteomics and bioinformatic analyses were performed on these samples to determine epigenetic changes to genes and effects of DNA methylation on protein expression. Differentially methylated regions and protein abundances were found in PI fetuses compared to controls affecting pathways associated with compromised immune, neural, cardiac, and bone development.

## 2. Materials and Methods

### 2.1. Animals, Viral Infections, and Fetal Collections

All animal experiments were approved by the Institutional Animal Care and Use Committee 08-16A-01 (11/01/09) at Colorado State University and by the Colorado State University Biosafety Committee (BSL1 and BSL2 approval 19-037B for BVDV). Viral infections and fetal sample collections are described in previous studies [23,29]. Briefly, a sample size of 3–4 animals per treatment group was determined to be sufficient to achieve a power of 1 using Lenth’s power calculator. Eight unvaccinated, BVDV naïve yearling Hereford heifers were purchased and confirmed to be seronegative for BVDV1 and 2 by virus neutralization assay and ear-notch BVDV antigen capture ELISA [23,28]. The heifers were bred by artificial insemination and pregnancies were confirmed by ultrasound on days 35 and 70 post insemination. On day 75 of gestation, 4 randomly selected animals were intranasally inoculated with 2 mL of sham MEM media to generate control fetuses while the other four animals were inoculated with 2 mL ncp BVDV2 strain 96B2222 at 4.4 log_10_ TCID_50_/mL to generate PI fetuses. Infected and control animals were confirmed infected or noninfected, respectively, by RT-qPCR and serum neutralization assay throughout gestation [23].

Seven fetuses were collected (4 control, 3 PI) by cesarean section on gestational day 245. Originally, 4 PI fetuses were generated; however, one fetus was aborted for reasons unrelated to BVDV infection. Control fetuses were collected prior to the collection of PI fetuses to avoid any cross-contamination of infected tissues. Standard surgical practices were used, and sterile surgical packs were replaced between each animal. Splenic tissue was sampled from the middle of the organ to ensure the inclusion of both red and white pulps. Samples were dissected into halves; one half was snap frozen in liquid nitrogen and stored at −80 °C for later DNA and protein extraction for this study.

### 2.2. Reduced Representation Bisulfite Sequencing

One milligram of splenic tissue was homogenized and subjected to the Qiagen DNeasy Blood and Tissue Kit (Qiagen, Hilden, Germany) according to the manufacturer’s instructions for DNA extraction. Extracted DNA (~300 ng) was sent to Zymo Research (Irvine, CA, USA) for genome-wide classic reduced representation bisulfite sequencing (RRBS/methyl-seq). The following methods were provided by Zymo Research. Samples were digested with 30 units of *MspI* (NEB; Ipswich, MA, USA) and purified with DNA Clean & Concentrator-5 (Zymo Research; Irvine, CA, USA). Fragments were ligated to preannealed adapters with cytosine replaced with 5′-methyl-cytosine according to Illumina’s guidelines. Ligated fragments greater than 50 base pairs were recovered with the DNA Clean & Concentrator-5 (Zymo Research; Irvine, CA, USA) then bisulfite treated with the EZ DNA Methylation-Lightning Kit (Zymo Research; Irvine, CA, USA). Samples were subjected to PCR with Illumina indices and products were then purified with DNA Clean & Concentrator-5 (Zymo Research; Irvine, CA, USA). Size and concentrations were confirmed with the Agilent 2200 TapeStation and libraries were sequenced on an Illumina platform.

### 2.3. Methylation Bioinformatics and Pathway Analysis

Raw BAM files received from Zymo Research were analyzed in R using the methylKit R package and the Colorado State University’s CVMRIT03 server, which identified differentially methylated regions (DMRs) [32]. DMRs were determined significant by Fisher’s exact test. DMRs were considered significant with *p* < 0.05 and a 25% or greater difference in methylation patterns. Gene IDs were identified using the genomation R package [33]. Quality control plots and gene ontology plots were generated with clusterProfiler, pathview, and gage R packages [34,35,36]. Bioinformatic code is available online [37]. Raw files are available in the NCBI GEO Database (Accession GSE188977). Pathway analysis for both methylation and protein data was performed using Ingenuity Pathway Analysis (IPA; Qiagen, Hilden, Germany).

### 2.4. Protein Sample Preparation

One hundred milligrams of frozen splenic tissue from day 245 fetuses was subjected to protein extraction via RIPA Lysis and Extraction Buffer (Thermo Scientific; Waltham, MA, USA) and Halt Protease and Phosphatase Inhibitor Cocktail (10 μL HALT/1 mL RIPA; Thermo Scientific, Waltham, MA, USA) according to the manufacturer’s instructions. Protein was quantified using a Pierce BCA Assay (Thermo Scientific; Waltham, MA, USA). Detergents were removed from the protein samples using the Pierce Detergent Removal Spin Columns (Thermo Scientific; Waltham, MA, USA). Cleaned samples were submitted to the CSU Analytical Resources Core Bioanalysis and Omics for protein quantification, trypsin digestion, and protein identification via liquid chromatography and mass spectrometry (LC-MS). The following methods were provided by the CSU Analytical Resources Core Bioanalysis and Omics. Aliquots of samples were diluted 1:2 in 2 M urea, 2% SDS and measured with the Pierce BCA Protein Assay (Thermo Scientific; Waltham, MA, USA). Fifty micrograms total protein was aliquoted and processed for in-solution trypsin digestion as previously described [38]. Absorbance was measured at 205 nm on a NanoDrop (Thermo Scientific; Waltham, MA, USA) and total peptide concentration was subsequently calculated using an extinction coefficient of 31 [39].

### 2.5. Liquid Chromatography and Mass Spectrometry

The following methods were provided by the CSU Analytical Resources Core Bioanalysis and Omics. Reverse phase chromatography was performed using water with 0.1% formic acid (A) and acetonitrile with 0.1% formic acid (B). One micrograms of peptides was purified and concentrated using an on-line enrichment column (Waters Symmetry Trap C18 100 Å, 5 μm, 180 μm ID × 20 mm column). Chromatographic separation was performed on a reverse phase nanospray column (Waters, Peptide BEH C18; 1.7 μm, 75 μm ID × 150 mm column, 45 °C) using a 90 min gradient: 5–30% B over 85 min followed by 30–45% B over 5 min at a flow rate of 350 nL/min. Peptides were eluted directly into the mass spectrometer (Orbitrap Velos Pro, Thermo Scientific; Waltham, MA, USA) equipped with a Nanospray Flex ion source (Thermo Scientific; Waltham, MA, USA) and spectra were collected over an *m*/*z* range of 400–2000 under positive mode ionization. Ions with charge state +2 or +3 were accepted for MS/MS using a dynamic exclusion limit of 2 MS/MS spectra of a given *m*/*z* value for 30 s (exclusion duration of 90 s). The instrument was operated in FT mode for MS detection (resolution of 60,000) and ion trap mode for MS/MS detection with a normalized collision energy set to 35%. Compound lists of the resulting spectra were generated using Xcalibur 3.0 software (Thermo Scientific; Waltham, MA, USA) with an S/N threshold of 1.5 and 1 scan/group.

### 2.6. Proteomics Data Analysis and Instrument Quality Control

The following methods were provided by the CSU Analytical Resources Core Bioanalysis and Omics. Tandem mass spectra were extracted, charge state deconvoluted, and deisotoped by ProteoWizard MsConvert (version 3.0). Spectra from all samples were searched using Mascot (Matrix Science, London, UK; version 2.6.0) against reverse concatenated Uniprot_Bovine_rev_102819 and cRAP_rev_100518 databases (UP000009136, downloaded 28 October 2019, and cRAP, downloaded 5 October 2018, 75,997 total entries) assuming the digestion enzyme trypsin. Mascot was searched with a fragment ion mass tolerance of 0.80 Da and a parent ion tolerance of 20 PPM. Carboxymethylation of cysteine was specified in Mascot as a fixed modification. Deamidation of asparagine and glutamine and oxidation of methionine were specified in Mascot as variable modifications.

Search results from all samples were imported and combined using the probabilistic protein identification algorithms implemented in the Scaffold software (version 4.11.1, Proteome Software Inc., Portland, OR, USA) [40,41]. Peptide thresholds were set (0.1% FDR) such that a peptide FDR of 0.05% was achieved based on hits to the reverse database [42]. Protein identifications were accepted if they could be established at less than 1% FDR and contained at least two identified peptides. Protein probabilities were assigned by the Protein Prophet algorithm [43]. Proteins that contained similar peptides and could not be differentiated based on MS/MS analysis alone were grouped to satisfy the principles of parsimony.

Relative quantitation was determined using spectral counting (SpC) [44]. A Student’s *t*-test was applied to determine protein species that were significantly different in abundance between groups (*p*-value < 0.05). Pseudo values were added (+1) prior to fold change calculations to eliminate zero values.

Instrument suitability was monitored through analysis of commercially purchased BSA standard digest and automated monitoring using PanormaQC. Metrics (e.g., mass accuracy, peak area, retention time, etc.) were monitored and flagged as outliers if results were outside +/− 3 standard deviations of the guide set (i.e., optimal operation). Values for all metrics were within normal limits throughout the duration of the experiment, indicating instrument stability and data robustness.

## 3. Results

### 3.1. Reduced Representation Bisulfite Sequencing

Classic RRBS indicated no significant changes in whole genome global methylation levels between control and PI animals. At the regional level, there were 2641 DMRs between control and PI animals. Of those DMRs, 1951 regions were hypermethylated in PI animals compared to controls while 691 regions were hypomethylated in PIs compared to controls. Due to the expansive results of the methyl-seq, results will be focused on significantly impacted pathways. All DMRs can be found in Appendix A.

The pathways reported are predicted to be downregulated, due to hypermethylation of genes, unless otherwise stated. Several pathways associated with general biological signaling were shown to be significantly affected, as predicted by IPA. In order of higher significance to lower significance, these pathways include but are not limited to: signaling by Rho family GTPases, human embryonic stem cell pluripotency, *Wnt/b-Catenin* signaling, protein kinase A signaling, *PI3K/AKT* signaling, calcium signaling (predicted upregulation), *STAT3* signaling, *mTOR* signaling, *phospholipase C* signaling, and *Wnt-Ca* signaling. Pathways associated with immune development and activation include the role of macrophages, fibroblasts, and endothelial cells in rheumatoid arthritis, *CCR5* signaling in macrophages (predicted upregulation), *IL8* signaling, **IL1*5* production, regulation of *IL2* expression in activated and anergic T lymphocytes, *IL1* signaling, *B cell activating factor* signaling, *CXCR4* signaling, *PKCθ* signaling in T lymphocytes, leukocyte extravasation signaling, and *CD27* signaling in lymphocytes (predicted upregulation). Pathways associated with neural development include axonal guidance signaling, synaptogenesis signaling pathway (predicted upregulation), *CREB* signaling in neurons, alpha adrenergic signaling, and synaptic long-term depression. Pathways associated with cardiac development include factors promoting cardiogenesis in vertebrates, role of *NFAT* in cardiac hypertrophy (predicted upregulation), and cardiac hypertrophy signaling. Pathways associated with bone development or bone disease include the role of osteoblasts, osteoclasts and chondrocytes in rheumatoid arthritis, osteoarthritis pathway, and actin cytoskeleton signaling. When considering both hypo- and hypermethylated regions of PI fetuses, the *nuclear factor of activated T cells* (*NFAT*) family and signaling pathway stood out. Within that signaling cascade, *calcium channel ORAI*, *calmodulin 1* (*CALM1*), and *NFATc2* were hypomethylated while co-stimulatory molecules *CD247*, *VAV1*, *NFATc1*, and *NFATc4* were hypermethylated in PIs compared to controls. Significantly different genes associated with these pathways can be found in Appendix A.

### 3.2. Proteomics

Proteomic analysis from LC-MS identified 12 significantly different proteins, with a 1.5-fold cutoff, in PI fetuses vs. controls. Increased proteins were translation machinery associated protein 7 (TMA7), peptidase d (PEPD), nuclear migration protein nudc (NUDC), tropomodulin 3 (TMOD3), and small nuclear ribonucleoprotein F (SNRPF). Proteins decreased in PI fetuses compared to controls include Thy-1 cell surface antigen (THY1), heterogenous nuclear ribonucleoprotein c (HNRPC), cysteine and glycine rich protein 1 (CSRP1), vesicle associated membrane protein associated protein B (VAPB), cystatin B (CSTB), AKAP2 c domain containing protein (AKAP2), and calponin 2 (CNN2).

## 4. Discussion

Day 245 fetal spleen samples are an imperfect but unique sample set for this study. As the developing fetus has a constantly changing epigenome, the fetuses on day 245, as opposed to early timepoints, provide epigenetic data most similar to what we would expect to see in postnatal animals. When discussing methylation data, it is important to understand its limitations without RNA expression data. Unfortunately, the data presented in this study do not contain RNA expression due to RNA from splenic tissue degrading over time at −80 °C; however, this leaves opportunities for future studies. Additionally, the epigenetic changes found in PI fetal spleens were not completely reflected in protein abundance. However, changes in methylation are rarely reflected in gene expression at one given time. Changes in gene expression due to methylation of specific genes may have occurred prior to day 245 or may occur postnatally, as discussed below. Despite these limitations to interpretation, the data discussed below are still highly valuable in understanding the effects BVDV PI has on fetal development.

Fetal BVDV PI spleens contained 2641 DMRs and differences in the expression of 12 proteins compared to controls. Hyper- and hypomethylated genes were found in pathways related to the immune system, osteoclastogenesis in bone, and the developing neural system and heart in PI fetal spleens.

### 4.1. Fetal BVDV PI Alters the Expression of Genes That Influence Lymphocyte Development

Previously, it had been hypothesized that the dramatic attenuation of BVDV PI fetal immune responses and, specifically, the inactivation/lack of response from lymphocytes were due to fetal Treg cells identifying BVDV as self [24]. This immunotolerance was expected to affect lymphocyte development and reaction to pathogens. The data presented here suggest that epigenetic changes affecting genes responsible for bone development may also interfere with hematopoiesis and cell trafficking with subsequent effects on lymphocyte and immune system development. Methylated pathways associated with immune system development are summarized in Figure 1.

*Interleukin 2* (*IL2*) is an important cytokine in the development and activation of T cells promoting both the activation and anergy of T cells as well as Treg development. Genes within the *IL2* pathways involved in the regulation of *IL2* action and *PKC* signaling in T cells were heavily hypermethylated. For example, *CD247* (also known as *CD3zeta*), part of a critical T cell receptor co-stimulatory molecule, was hypermethylated. Reduced expression of *CD247* would reduce the release of *IL2*. *IL2* is a potent stimulator of *FOXP3* expression in Treg development, potentially causing an increase in Treg cell development through increased *FOXP3* expression [45,46]. Treg cells compete with T helper (Th) cells for available *IL2*, causing apoptosis of effector cells [45,47]. *IL2* knockout mice exhibit lymphoproliferation and lethal autoimmunity which is prevented by adoptive transfer of normal Treg cells [48,49]. In a previous study, it was hypothesized that PI fetuses develop normal Treg cells prior to/during BVDV infection prior to the development of antigen-specific T cells, causing Treg cells to identify viral antigens as self [24]. With the current data, it is postulated that as the fetal spleen and thymus develop, genes in the *IL2* signaling pathway of *CD4* Th cells are hypermethylated (*CD247*, *VAV*, *BMP3*, and *SMAD3*), shifting the *IL2* signaling towards the maintenance of Treg cells while hindering Th cell development. Methylation of these genes was not reflected in proteomic data. Thus, the changes in protein levels may have occurred earlier or later in gestation or postnatally.

Leukocyte extravasation is the movement of cells from the blood stream across blood vessel walls to infected tissues in response to release of cytokines and chemokines as a part of the innate immune response. Recruitment of leukocytes requires gene products that stimulate leukocyte migration and adhesion to endothelial cells (ECs). Several signaling genes within this pathway were hypermethylated in fetal PI spleens including *vav guanine nucleotide exchange factor 1* (*VAV1*), *integrin subunit alpha M* (*ITGAM*), *THY1*, and others. The effects of hypermethylation of *THY1* were reflected in decreased THY1 protein in PI fetal spleens compared to controls. *THY1* is expressed in fibroblasts, neurons, and hematopoietic stem cells and has several functions including mediating the binding of leukocytes to ECs and triggering neutrophil effector functions. *THY1* knockout mice have decreased extravasated leukocytes and altered cytokines released at sites of inflammation [50]. TMOD, an increased protein in PI fetal spleens, has been shown to negatively regulate endothelial cell motility by capping pointed ends of actin structures possibly contributing to decreased leukocyte migration [51]. Despite *TMOD* not being differentially methylated, other mediators involved in the pathway have affected its expression.

*C-X-C motif chemokine receptor 4* (*CXCR4*) was previously shown to be decreased for several months in maternal peripheral blood mononuclear cells (PBMCs) in the dams carrying PI fetuses and in postnatal PI animals compared to control animals [21]. Moreover, CXCR4 levels in herds with several PIs were decreased compared to herds without PI animals [52]. CXCR4 has several functions in regulating leukocyte trafficking, lymph node organization, T cell priming, B cell development, and bone marrow homeostasis, and is essential for normal cardiac development [53]. Although *CXCR4* was not differentially methylated or expressed as a protein, its signaling pathways, including G proteins and Rho signaling, were hypermethylated in the data from the current experiment [53]. Inhibition/hypermethylation of G protein and Rho signaling may give insight into the *CXCR4* inhibition in PI animals, possibly contributing to dysfunctional leukocyte and lymphocyte development in PI animals. These genes were not reflected in the proteomics data, indicating that their differential expression may occur at an earlier time point during fetal development.

AKAP2 protein was decreased in fetal BVDV PI spleens compared to controls. AKAP family members anchor and compartmentalize PKA for regulation of cAMP signaling [54]. More recently, AKAP2 was identified as an upregulated protein in CD8 cells responding to *IL35* stimulation, suggesting a role for AKAP2 in the differentiation of CD8 T cells [55]. AKAP2 was also increased in T cells from patients with systemic lupus erythematosus, suggesting a role for AKAP2 in Treg cell functions [56]. The literature for AKAP2 is limiting and the reason for its decrease is unknown; however, there was no change in the methylation of the gene, possibly due to mRNA half-life or stability. Its decrease in BVDV PI fetal spleens suggests a role in immune dysfunction and dysfunction of the PKA/cAMP signaling pathways.

CNN2 has been associated with the development of immune cells of myeloid lineage. Peripheral blood neutrophils and monocytes were decreased in CNN2 interrupted mice, concurrently with increased proliferation and migration of neutrophils and monocytes, and increased macrophage phagocytic activity. In contrast, increased CNN2 was associated with the adhesion dependent maturation of macrophages. In this study, CNN2 was decreased in BVDV PI fetal spleens, which could cause a decrease in mature macrophages/leukocytes while increasing the proliferation and migration of immature monocytes in PI animals. These effects could be due to direct effects on macrophage development, or due to osteopetrosis resulting in the migration of immature leukocytes from bone marrow to the spleen and increased extramedullary splenic hematopoiesis.

### 4.2. BVDV Fetal Infection Alters Osteoclastogenesis Resulting in Extramedullary Splenic Hematopoiesis

Three hundred and eighty-five million years ago, the vertebrate skeletal system and immune system evolved together as aquatic vertebrates became terrestrial [57]. The evolution of the acquired immune system and the skeletal system together has been theorized as the reason for the interplay between the two systems [57,58]. Several of the same molecules that affect bone development also affect immune development/activation in modern mammals [57,59,60]. In the early 2000s, the term “osteoimmunology” was coined to accurately describe the importance of the two systems and their interactions [57,59,60]. In addition to providing support for locomotion and minerals for homeostasis, the bone is a primary lymphoid organ, in which hematopoietic stem cells, lymphocytes, monocytes, and macrophages develop and reside [61]. The bone marrow houses chondrocytes, osteocytes, osteoblasts, and osteoclasts critical for bone metabolism and is constantly being remodeled, built by osteoblasts and resorbed by osteoclasts [61]. The progenitors of osteoclasts are macrophages/monocytes of hematopoietic origin, and their differentiation is mediated by *macrophage colony stimulating factor* (*CSF1* or *M-CSF*), *receptor activator of nuclear factor kappa B ligand* (*RANKL*), and others [61]. Although mediated by *CSF1* and *RANKL*, *NFATc1* is considered the master transcription factor for osteoclastogenesis [61,62,63,64]. Osteoclast bone resorption is also stimulated by the inflammatory interleukins, *IL1* and *IL6*, secreted by antigen stimulated immune cells [61]. Osteopetrosis, also known as marble bone disease, is associated with approximately 27 human genetic mutations, and characterized by brittle bones and reduced size of the bone marrow cavity [57,65,66]. Postulated to be caused by a decrease in osteoclasts, the reduction in the size of the bone marrow cavity severely restricts hematopoiesis in this location, forcing the movement of hematopoietic cells to lymph nodes, the spleen, and/or the liver. Extramedullary hematopoiesis results in severe anemia and impairs immune cell differentiation [57,65,66]. Previous studies reported osteopetrosis and abnormal bone development in BVDV PI animals and in border disease virus infected lambs [17,18,20,26,28]. Three of six PI fetuses exhibited thickened femoral cortical bone and a significantly smaller medullary space compared to controls, and day 245 BVDV PI fetuses had transverse bands and lesions in the middle and distal zones of the tibia and femur, indicative of osteopetrosis and altered osteoclast differentiation/osteoclastogenesis [20,28]. Decreased numbers of osteoclasts and osteoblasts were observed in the PI tibias, further evidence of abnormal osteoclastogenesis and bone development [20]. In a case study, postnatal PI calves were observed to have irregular bands and lesions in the femur and tibia [17]. The calves lacked osteoclasts and megakaryocytes in the bone marrow, which also exhibited smaller bone marrow spaces [17]. Additionally, the calves had lymphoid depletion in the spleen and lymph nodes [17]. Increased hematopoietic activity was observed in fetal PI spleens early in gestation, while the day 245 spleens exhibited fewer lymphocytes [24]. Another BVDV case study revealed dense bones with striped appearance in a two-day old calf admitted for a femur break [18]. At 13 months of age, necropsy revealed lymph node hyperplasia, extramedullary hematopoiesis in the spleen, and BVDV infected osteoblasts, osteocytes, and splenic blood leukocytes [18]. All three studies agree that the observed changes are the result of decreased osteoclast numbers and suggest different mechanisms for osteopetrosis, including BVDV induced secretion of *IL1* inhibitor from monocytes, direct viral changes to bone marrow derived cells, and inhibition of osteoclastogenesis by type I IFNs [57,65,66]. Similar bone lesions and pathologies are found in human neonates following transplacental infection with human cytomegalovirus (CMV) and rubella virus, characterized by irregular radiodense zones and pathologic fractures [26,67,68,69,70,71].

Several pathways identified by IPA including signaling by Rho family GTPases, *Wnt/b-Catenin* signaling, *protein kinase A* signaling, *PI3K/AKT* signaling, calcium signaling, *STAT3* signaling, phospholipase C signaling, and *Wnt-Ca* signaling pathways contained the same differentially methylated genes affecting osteoclastogenesis in PI fetal spleens compared to controls. Changes in signaling due to DNA methylation may explain the decreased number of osteoclasts observed in PI animals (summarized in Figure 2) [57,65,66]. The inhibition of osteoclastogenesis would dramatically reduce calcium levels in the body, as bone resorption normally maintains extra- and intracellular calcium levels [72]. Most genes in the signaling pathway of the Rho family GTPases are hypermethylated, affecting *activator protein transcriptional complex* (*AP1*) formation/AP1 gene expression. The *AP1* transcription complex includes the cFos and Jun family proteins. The cFos proteins are stimulated by not only Rho family GTPases, but also *nuclear factor kappa B* (*NFKB*). *NFKB* is expected to be inhibited due to hypomethylation of inhibitor of *NFKB* subunit beta and *peroxisome proliferator activated receptor gamma* (*PPARG*), and hypermethylation of *PPARG coactivator 1 beta* (*PGCA1B*), and *CCAAT/enhancer binding protein alpha* (*C/EBP1A*) [61,73,74,75,76]. As a result of the increased methylation of Rho family GTPase pathway and cFos stimulants, the cFos and *AP1* complexes are thought to be decreased and, thus, unable to stimulate osteoclastogenesis. In mouse studies, mice lacking cFos proteins developed osteopetrosis due to impaired osteoclastogenesis [61,73,77].

The *Wnt* signaling pathway is important for both osteoclast and osteoblast differentiation, and the role of Wnt signaling depends on which *Wnt* molecule binds to the Frizzled receptor (hypermethylated). Normally, *Wnt2b* expression is higher in mature osteoblasts compared to osteoblast progenitors, however, its direct role in osteoblast differentiation is unknown. *Wnt2b* was hypermethylated in our data, possibly inhibiting not only its signaling pathway, but also affecting osteoblast maturation [78]. *Wnt7a* was also hypermethylated in our data. Previous studies have shown *Wnt7* conditional knockout causes a decrease in bone formation and chondrocyte differentiation [79]. The methylation of this pathway is intriguing as most of the genes are hypermethylated; however, inhibitors of the pathway are also hypermethylated. The predicted decreased expression of hypermethylated genes in the pathway is expected to cause altered bone development and hindered osteoblast and osteoclast maturation/differentiation, contributing to the osteopetrotic pathology and decreased osteoclast numbers seen in these fetuses [20,28]. Osteoblast derived *dickkopf WNT signaling pathway inhibitor 1* (*DKK1*), a *Wnt* inhibitor, is also hypermethylated which may compensate for a decreased osteoclastogenesis. *Phospholipase C* signaling contains the same signals as the Rho family GTPases pathway with direct effects on *nuclear factor of activated T cells* (*NFATc1*) expression. The hypermethylation of the G proteins and Rho is expected to decrease the signal for calcium mediated gene expression.

The pathways significantly affected by methylation, identified by IPA, which were associated with osteoarthritis and bone development, had overlapping genes and signaling pathways, with the *Wnt* signaling pathway being present in all bone related pathways. *Colony stimulating factor 1* (*CSF1*; also known as macrophage colony stimulating factor) is hypermethylated and a large inducer of osteoclastogenesis, macrophage differentiation, and osteoclast survival. *CSF1* is produced by osteoblasts and activated T cells to induce differentiation of macrophages or dendritic cells into osteoclasts [80]. Therefore, hypermethylation of *CSF1* would greatly inhibit osteoclastogenesis from monocytic cells. Mice lacking *CSF1* exhibit severe osteopetrosis due to a decrease in osteoclastogenesis [81,82]. Interestingly, *IL1* and *IL6* receptors, both inducers of osteoclastogenesis, are hypomethylated. *IL6* is secreted by bone marrow stromal cells, osteoblasts, and macrophages to induce an inflammatory response as well as osteoclastogenesis through the *STAT3* (hypermethylated) pathway [83,84]. Due to the secretion of *IL6* from bone marrow derived cells, the actual expression of *IL6* may be hindered due to the osteopetrotic pathologies of BVDV PI animals lacking bone marrow and, thus, bone marrow derived cells. In response to such a decrease, **IL6*R* may be hypomethylated to amplify the signal for downstream targets if *IL6* does happen to bind. Despite the attempted amplification, the hypermethylation of *STAT3* would further hinder the *IL6* signaling for osteoclastogenesis. However, *IL6* is also known to be antiosteoclastogenic but, because *STAT3* is hypermethylated, the signaling will be hindered no matter the reason [84]. *IL1* is another proinflammatory cytokine that is pro-osteoclastogenic. One study has determined that BVDV PI animals have inhibited *IL1* expression through the expression of an *IL1* inhibitor from infected cells [85]. This inhibition of *IL1* may be an indirect cause of the hypomethylation of its receptor, for the same proposed reasons as the hypomethylation of **IL6*R* discussed above.

*NFATc1* has been identified as the master transcription factor for osteoclast differentiation [64]. In healthy mammals, *RANKL* induces calcium oscillations in the bone marrow cells, activates calcineurin, then induces *NFATc1* expression via *TNF receptor associated factor* (*TRAF*) 6 and cFos signaling [64]. *NFATc1* autoamplifies and, with the assistance of the *AP1* transcription complex, stimulates transcription of genes for osteoclastogenesis [64]. In the methylation data, *NFATc1* was both hypo- and hypermethylated. Although the interpretation of this data is difficult, differing methylation patterns on a gene have been associated with carcinogenesis [86]. One study suggests that in the case of differing methylation patterns on one gene, hypermethylation is associated with developmental process functions and regulation of biological processes, while hypomethylation is associated with a response to stimulus (inflammation, infection, etc.) [86]. Despite the lack of information on this specific phenomenon, it is clear that *NFATc1* expression would be altered and would cause an imbalance in osteoclast differentiation signaling. Interestingly, an upstream regulator to *NFATc1*, *calmodulin*, was hypomethylated. *Calmodulin* is a regulator for all *NFAT* family members and is involved with general calcium signaling. The hypomethylation of the *calmodulin* gene could be an attempt to stimulate other *NFAT* family members or amplify the calcium signaling pathway discussed above.

*CSTB* has several biological functions including chemotaxis, stimulation of cytokine secretion, release of nitric oxide, regulation of apoptosis, protection of neurons, cell cycle regulation, and bone resorption. *CSTB* was identified as a decreased protein in PI animals compared to controls. *CSTB* knockout mice mimic the progressive myoclonus epilepsy of Unverricht–Lundborg type (*EPM1*) epileptic phenotype in humans, characterized by inflammation in the brain [87]. In normal homeostasis, *CSTB* is also thought to protect osteoclasts from apoptosis in vitro through the inhibition of *cathepsin K* [88]. A normal hypothesis for *CSTB* knockouts would be an increase in osteoclasts, due to a lack of available inhibition of *cathepsin K*; however, that is not the case in vivo for *CSTB* deficient or knockout mice. In addition to epileptic phenotypes, *CSTB* knockout mice showed a significant decrease in osteoclasts and increased bone mineral densities [89]. A direct inhibition of *CSTB* has been shown in mice infected with ectromelia virus, which is hypothesized to enhance viral replication [90]. Similarly, the PI BVDV infection may be targeting *CSTB* in dendritic cells to increase its own replication and contribute to the inhibition of osteoclastogenesis.

VAPB is a vesicle associated protein which can form dimers with itself or VAPA for vesicle transport. VAPB protein was decreased in PI animals compared to controls. VAPB has been shown to enhance hepatitis C virus (another flavivirus) replication through the binding of the viral protein NS5B to intracellular vesicles. If VAPB also enhances BVDV replication, then the amount of VAPB protein would not be expected to be decreased and VAPB bound to BVDV would have been counted and possibly increased in PIs. Despite this potential link between BVDV replication and VAPB, the decreased VAPB protein is thought to be reflective of its function in osteoclastogenesis and cardiac development, or because it is directly targeted by the virus. VAPB has a role in osteoclastogenesis, by controlling the activation of the phospholipase C (PLCG)-Ca-NFATc1 signaling pathway [91]. With decreased VAPB, *NFATc1* decreases along with osteoclastogenesis [91]. The specific method by which VAPB regulates this pathway is still not known [91]. In humans, deficiency in VAPB results in amyotrophic lateral sclerosis (ALS) [92]. Through its association with ALS, VAPB was found to be required for neural and cardiac pacemaker channels. Further, mice deficient in VAPB present with bradycardia, suggesting the degeneration of sympathetic neurons, causing prolonged QT intervals [92]. BVDV antigen has been found in the Purkinje cells of the heart and it was speculated that the presence of the virus in the conduction system could cause arrhythmias [93]. Although cardiac arrhythmias have not been described in BVDV PI animals to date, it is possible that decreased VAPB could contribute to altered cardiac development, discussed below.

Although these epigenetic and proteomic data were procured from the fetal spleen, they most likely reflect pathologies observed in the bone. Hematopoietic stem cells are produced in the bone prior to migration into peripheral lymphoid organs such as the spleen [57,65,66]. If hematopoiesis is hindered in bone because of decreased bone resorption and a significantly smaller bone marrow cavity, hematopoietic stem cells would migrate to the spleen, resulting in extramedullary splenic hematopoiesis [57,65,66]. Thus, epigenetic changes and protein differences found in the PI fetal spleen are likely reflective of the effect of BVDV on osteoclastogenesis, bone, and bone marrow changes [18]. Additionally, cells derived from extramedullary splenic hematopoiesis are mobilized to peripheral organs such as the heart and brain, discussed below [94]. Alterations in the methylation patterns and protein abundance seen in the PI fetal spleen may have effects on said peripheral organs through the mobilization and migration of hematopoietic cells, further contributing to BVDV PI pathologies.

### 4.3. Methylation Patterns and Protein Expression in the PI Fetal Spleen Are Related to Neural Defects in PI Fetuses

BVDV directly infects fetal neuronal tissues and is associated with a variety of neural defects including cerebellar hypoplasia, hydranencephaly, hydrocephalus, and retinal dysplasia [27,95,96,97]. In PI cattle, BVDV antigen is present in up to 90% of neurons including dendrites and axons [97], and has been localized to neurons in the cerebral cortex, olfactory lobe, hippocampus, hypothalamus, corpus quadrigeminus, pons, medulla, cerebellum, and spinal cord [96]. Infection of the fetal brain occurs rapidly following maternal infection as BVDV positive microglial cells, oligodendrocytes, decreased oligodendrocyte precursor cells, reduced numbers of neuron specific enolase positive nerve cells, hypomyelination, and vascular lesions are observed as early as 22 days post maternal inoculation [27,95,97]. Although the pathologic lesions are well studied, the molecular mechanisms causing the defects have not been investigated.

The most significantly methylated canonical pathway identified by IPA was the axonal guidance signaling pathway with 60 differentially methylated genes, with 43 genes hypermethylated (72%), including G protein receptors, transcriptional regulators, enzymes, growth factors, kinases, and others. The affected genes/pathway are predictive of alterations in actin filament reorganization, microtubule assembly, axon attraction, and adhesion. In addition, genes within the synaptogenesis pathway were differentially methylated with most regions being hypermethylated. These genes are involved in neuron adhesion, microtubule stabilization, and axonal guidance signaling. In addition to the epigenetic changes, proteins VAPB, THY1, and CSTB (also known as EPM1), which were significantly decreased in PI fetal spleens, are associated with altered neural development. CSTB mutations are associated with EPM1 while VAPB genetic mutations have been associated with ALS in humans [87,98]. A study of VAPB and the ALS phenotype and found decreased VAPB protein in cells from ALS patients and motor neurons from ALS animal models contributing to motor neuron degeneration [99,100,101]. *THY1* is expressed on neurons and is an inhibitor of neurite outgrowth. While *THY1* knockout mice do not have structural neural abnormalities, they exhibit alterations in spatial learning and a lack of social cues, possibly reflecting abnormal neurite connections [102,103,104]. *CSTB* knockout mice exhibit alterations in GABAergic signaling, possibly contributing to neuronal degeneration [98]. The decreased expression of these hypermethylated genes predicts decreased neuron and axon growth, cell adhesion, and reduced formation of synapses which may explain some of the developmental defects of the fetal brain characteristic of BVDV PI fetal infections. The connection between DNA methylation and protein expression in fetal spleen and neural development is not known. The changes in DNA methylation of genes may be universal across fetal tissues, including the brain, and/or alterations in the concentration of proteins in the fetal circulation may influence the development the fetal brain.

### 4.4. Genes Affecting Cardiac Development Are Hypermethylated in the Spleen of BVDV PI Fetuses

IPA identified several heavily methylated pathways associated with cardiac development, summarized in Figure 3. These pathways include the previously described pathways *Wnt* signaling, G proteins, and calcium signaling. Several hypermethylated genes promoting the cardiogenesis in vertebrates pathway contribute to a dysfunctional cell cycle of cardiocytes and cardiomyocyte differentiation, including *transcription factors/T cell factors 4 and 7* (*TCF*), *t-box transcription factor 5 (TBX5*), *cyclin D1*, and *myocardin*. In the pathway for the role of *NFAT* in cardiac hypertrophy, *histone deacetylase 7* and *NFATc4* were both hypermethylated. Mice with attenuated *NFAT* (specific *NFAT* not identified) in late gestation showed a thinning of the myocardium [105]. The cardiac hypertrophy signaling pathway also identified the hypermethylated gene, *myocyte enhancer factor 2D* (*MEF2D*). Interestingly, *VEGF* was also hypermethylated. *VEGF* is needed in the developing fetus for the organization of the vascular system and its predicted decreased expression may alter said organization [106]. The methylation patterns of these pathways and genes suggest disruptions in cardiac development of the BVDV PI fetuses. Evidence for abnormal heart development in PI animals includes enlarged hearts and chronic passive congestion in aborted fetuses, and angioendotheliomatosis, myocardial necrosis, and fibrosis in postnatal PI animals [107,108].

### 4.5. BVDV Replication and Immune Evasion

Several of the increased proteins in PI animals, compared to controls, have a link to viral replication. Not much is known about *TMA7*; however, it has been shown to be increased in influenza viral infections [109]. In a STRING analysis, TMA7 interacts with NFATc2 interacting protein; however, the importance and function of this interaction is not known. PEPD has been shown to bind flaviviral NS5 to inhibit IFN receptor 2 and, thus, type I IFN signaling, as a way to evade the host immune response [110]. *NUDC* has a role in cell migration, hematopoiesis, and spindle formation, and as a regulator of inflammation [111,112]. Its roles are not well understood and its role in BVDV PI animals is difficult to speculate. *NUDC* may be contributing to PI immune evasion by regulating inflammatory responses or may be an indication of malignant proliferation of hematopoietic stem cells in the spleen.

BVDV viral proteins have been shown to inhibit and evade several immune functions in vitro, including IFN responses [31,113,114,115,116,117]. The authors suggested that BVDV only inhibits/evades a type I IFN response to itself directly but does not inhibit a response to any other invading pathogen [31,113,114,115,116,117]. Although in vivo models have shown a robust type I IFN response in PI fetuses following maternal infection, one cannot discount a possibility that BVDV inhibits IFN responses in vivo. Previous studies have shown a PI fetal IFN response at day 97 of gestation, followed by a drastic downregulation of both the innate and adaptive immune responses by day 190 of gestation, seen in mRNA expression data [24,25]. Interestingly, when comparing a previous microarray from day 97 fetal spleens to our current day 245 methylation data, we found an inverse relationship between day 97 mRNA differential expression and day 245 methylation/protein [24]. Of particular interest are *NFATc2* (decreased mRNA on day 97, hypomethylated by day 245), *CD247* (increased mRNA on day 97, hypermethylated by day 245), *VAV1* (increased in day 97 mRNA, hypermethylated by day 245), *ITGAM* (increased in day 97 mRNA, hypermethylated by day 245), and *CSTB* (increased mRNA on day 97, decreased protein by day 245) [24]. To our knowledge, time course studies with mRNA, methylation, and protein data have not been done previously, therefore we can only speculate on the reason for this inverse relationship, with several possible explanations. The first explanation is that day 97 PI fetuses exhibited a peak of immune response, whether it was a direct response to the virus or stimulated by placental inflammation. In response to this drastic increase in the fetal immune response, some of the stimulated immune genes may be differentially methylated between days 97 and 245, overcorrecting for the fetal immune response on day 97. The second explanation may be that as Treg cells identify BVDV as “self”, Treg cells shut down the stimulated genes and in an attempt to prevent further response to the virus, the fetus differentially methylates the stimulated or downregulated genes. The third explanation is that the PI fetal immune response on day 97 is only a reaction to maternal and placental inflammation. Once maternal or placental inflammation is subdued, BVDV evades and possible downregulates the immature fetal immune response as suggested by in vitro studies, causing differential methylation patterns later in gestation. All of these explanations are reflected in the downregulation of day 190 PI fetal spleen immune related mRNAs measured in previous studies [24,25]. These day 190 mRNAs were specific targets measured via RT-qPCR and were not differentially methylated; however, their differential expression may be a direct result of the extramedullary splenic hematopoiesis previously seen in other studies and suggested by the current study [18].

## 5. Conclusions

Methylation of genes and the impact on protein expression may help explain the pathologies in multiple organ systems associated with early BVDV transplacental infections. We suggest that the inflammation and cytokines associated with the early fetal response to BVDV infection, along with Treg suppression of the immune system during immunotolerance, inhibit proper osteoclastogenesis and bone formation (summarized in Figure 4). In earlier studies, our group found an upregulation of *IFNB*, *IFNG*, and *ISG15* in day 97 PI fetuses (21 days post maternal infection) [24,29]. These IFNs and *ISG15* are known inhibitors of osteoclastogenesis, and their expression of these IFNs early in fetal development may not only inhibit ongoing osteoclastogenesis, but also cause the inhibition of osteoclastogenesis through hypermethylation of genes in most pathways affecting bone development and immune cell development [57,118,119]. By day 190 of gestation, Treg cells identified BVDV as self and suppressed the immune response to viral antigens; however, the epigenetic damage would have already been done. The results presented correlate with previously described pathologies observed in PI cattle postnatally, suggesting that the epigenetic changes caused by BVDV fetal infections have a large role in fetal development.

BVDV is an important economic problem for cattle industries worldwide. While this research enhances our understanding of viral repercussions in PI animals, it also provides insight into potential epigenetic changes due to maternal and fetal viral infections in humans. Maternal infection with Zika virus (ZIKV) and other viruses causes severe defects of the fetal brain [120]. Osteopetrosis has been identified in infants positive for CMV with abnormal amounts of leukocytes [121,122]. Additionally, children from mothers infected with rubella virus or CMV early in gestation had an increased risk of congenital heart defects [123]. While developmental defects of the brain, heart, and limbs may be readily diagnosed during the neonatal period, immune system dysfunctions would escape detection in human infants. This study provides potential explanations for several pathologies described in BVDV PI cattle, identifies possible targets for therapeutic treatments, and provides an intriguing model for the repercussions of transplacental viral infections.

Maternal factors contributing to fetal programming and health as an adult are a concept described as the developmental origin of adult health and disease; originally a hypothesis by Barker and Osmond (1986) based on the effects of maternal heart disease on fetal growth [124,125]. Fetal development is sensitive to the maternal environment, including maternal diet, body mass index, mental health (anxiety, stress, depression) and bacterial/helminth/viral infections, and inflammation. These factors not only affect fetal growth, but the epigenetic programming of genes [124,126]. In humans, the transplacental transmission of pathogens such as *Toxoplasma gondii*, HIV, parvovirus, *Listeria monocytogenes*, *Treponema pallidum*, varicella zoster virus, rubella virus, CMV, herpesviruses (HSVs) 1 and 2, and ZIKV causes significant detrimental effects on the fetus [127]. Although BVDV is a virus that affects ruminants and pigs only, the ability of BVDV to cross the placenta and affect the development of multiple fetal organ systems presents a unique opportunity to dissect the cellular and immune mechanisms by which pathogens affect gene expression with consequences for fetal development. These mechanisms and principles may give insight into the effects of human maternal infection on fetal development.

## Figures and Tables

**Figure 1 viruses-14-00506-f001:**
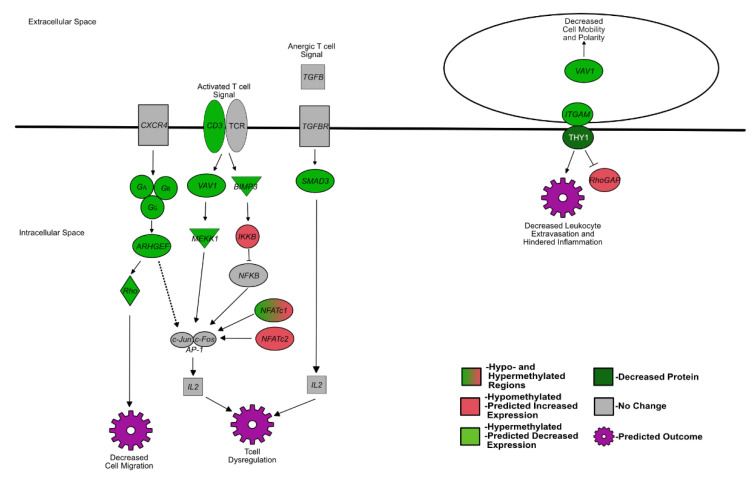
Summarized differentially methylated pathways and genes in PI fetal spleens associated with development of the fetal immune system. Green = hypermethylated, predicted decreased expression; red = hypomethylated, predicted increased expression; gray = not differentially methylated or not identified in RRBS; white text = protein.

**Figure 2 viruses-14-00506-f002:**
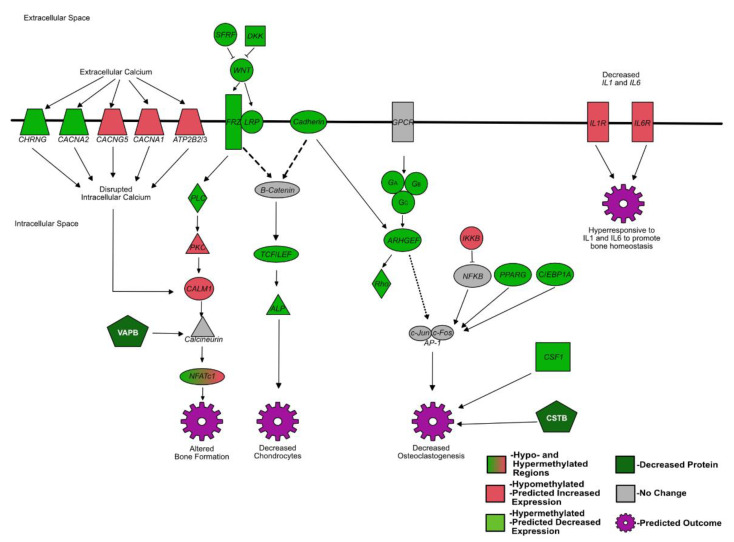
Summarized differentially methylated pathways in PI fetal spleens associated with bone development and osteoclast differentiation. Green = hypermethylated, predicted decreased expression; red = hypomethylated, predicted increased expression; gray = not differentially methylated or not identified in RRBS; white text: protein.

**Figure 3 viruses-14-00506-f003:**
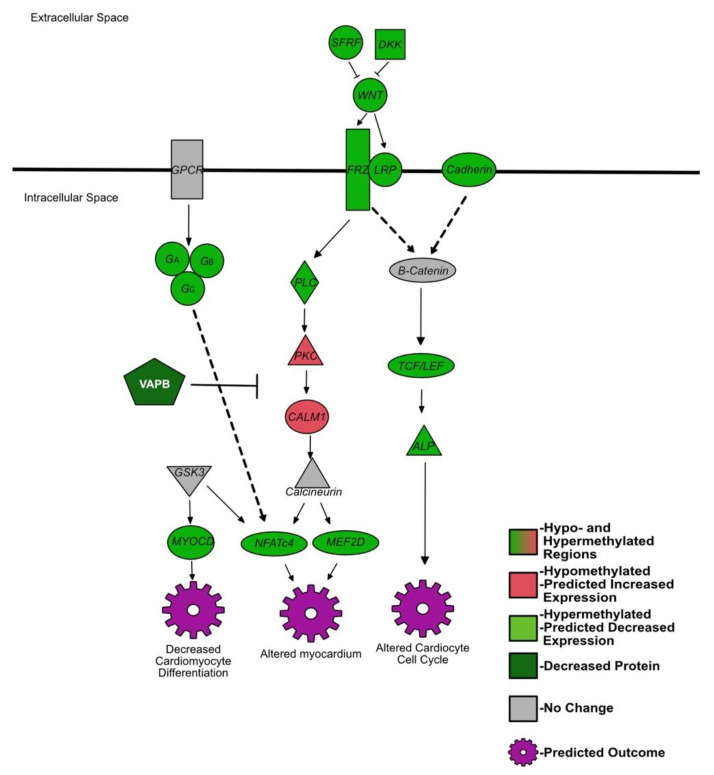
Summarized differentially methylated pathways and genes in PI fetal spleens associated with cardiac development. Green = hypermethylated, predicted decreased expression; red = hypomethylated, predicted increased expression; gray = not differentially methylated or not identified in RRBS; white text = protein.

**Figure 4 viruses-14-00506-f004:**
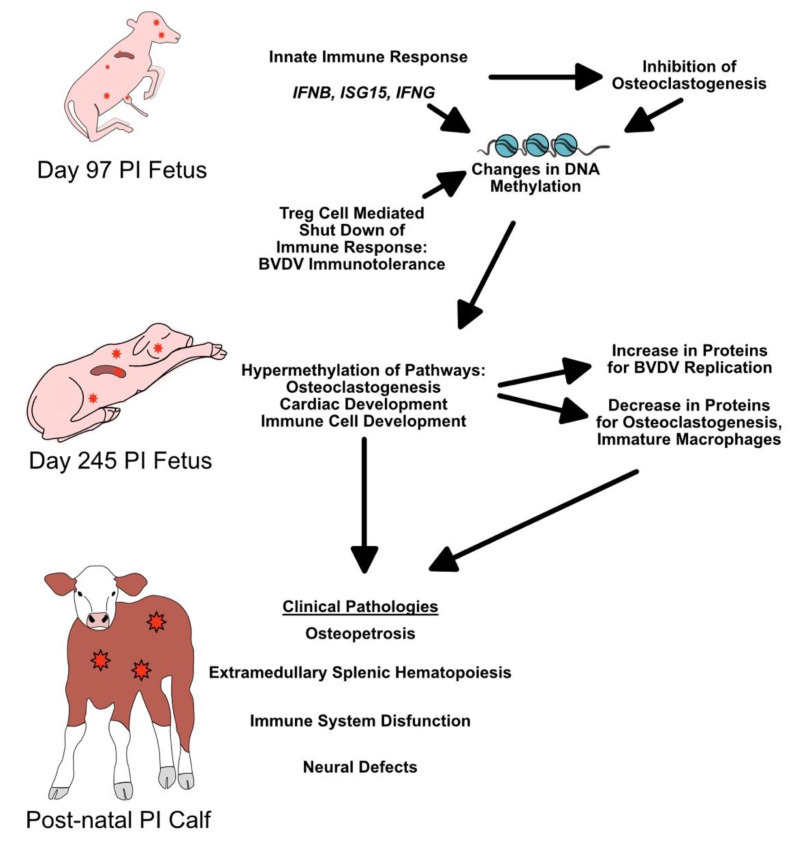
Hypothesized changes in prenatal PI calves in response to BVDV fetal infection and its effects on postnatal calf pathologies. Red stars = BVDV antigen. Day 97 data from Georges et al. 2020 and Smirnova et al. 2012.

## Data Availability

Raw data files are available in the NCBI GEO database. Accession numbers and reviewer access codes are listed in the methods.

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
