# Peer review of "Epigenomic and Proteomic Changes in Fetal Spleens Persistently Infected with Bovine Viral Diarrhea Virus: Repercussions for the Developing Immune System, Bone, Brain, and Heart"

_viruses, 2022, doi:10.3390/v14030506_

Round 1

Reviewer 1 Report

This manuscript described differences in gene expression in the spleens of BVDVPI and non-infected fetuses, with particular emphasis on detecting differences in the expression of molecules involved in the immune response. It would have been a more advanced report if similar data had been analyzed in thymus and lymph nodes in PI fetuses. This article is an interesting and valuable report because it provides a lot of indirect evidence for immune tolerance. I consider it matches the interest of the readers.

Author Response

Introductory Remarks

We would like to graciously thank the reviewers for their quick and thorough review of our paper. We have made appropriate edits as suggested and responses are below.

Reviewer 1 Comment: “It would have been a more advanced report if similar data had been analyzed in thymus and lymph nodes in PI fetuses.”

Response: Thank you for your review. We agree that thymus and lymph node data would have provided a much more robust experiment and report. Unfortunately, due to limited samples from our in vivo model and the expense when completing global methylation studies, we were not able to investigate methylation patterns of thymus or lymph nodes.

Reviewer 2 Report

The authors demonstrated that fetal BVDV infection resulted in hypermethylation of DNA associated with immune system, neural, cardiac, and bone development and down-regulation of protein expression associated with lymphocyte migration and development. These findings also provides insight into potential epigenetic changes due to maternal and fetal viral infections in humans and merit publication. For the benefit of the reader, however, data presentation should be improved using ex. PANTHER  classification system.

Minor points:

  1. Line 32. BVDV should be spelled-out.
  2. Line 51 and others. In vitro and in vivo should be in Italic.
  3. Line 108. MspI not Msp1.
  4. Lines 120-121. This sentence should be revised.
  5. Line 156. nL not nanoliters.
  6. Line 187. Student's should start with a capital letter.
  7. Lines 201-202. Delete differentially methylated regions ().
  8. Line 203 and others. Phospholipase C not phospholipase c.
  9. Line 289. Th not T helper.
  10. Line 402 and others. Rho should start with a capital letter.
  11. Lines 454-455. Delete myoclonus epilepsy of Unverricht-Lundborg type ().
  12. Lines 455-456. Delete amyotrophic lateral sclerosis ().
  13. Line 583. Influenza virus infection.
  14. Line 637. IFNs not interferons.
  15. Line 649. Zika virus (ZIKV).
  16. Lines 650-652. CMV not cytomegalovirus.
  17. Line 668. ZIKV. 

Author Response

Introductory Remarks

We would like to graciously thank the reviewers for their quick and thorough review of our paper. We have made appropriate edits as suggested and responses are below.

Reviewer 2

We are very grateful for your thorough review, especially the corrected acronyms in your specific comments below. Suggested edits have been made in the text.

Comment: For the benefit of the reader, however, data presentation should be improved using ex. PANTHER  classification system.

Response: We do agree that pathway and classification analyses such as PANTHER, Gene Ontology, and IPA increase the understanding of affected biological systems with listed proteins, nucleic acids, methylation, etc. Methylation and protein data were analyzed using IPA (Figures 1-3) which was very helpful in the understanding of biological processes involved in this model. However, the bovine species is not well represented in these analytical tools and results are limited. IPA was the most robust tool for our model and was chosen for our analyses. In Figures 1-3, proteins within those pathways were included in the figures. In response to your comment, we have added text to the protein methods to further clarify this. Upon your suggestion, we did use PANTHER; however, the results did not further clarify protein functions from what was given by IPA. Due to limited significant proteins, we did not pursue reporting PANTHER results, but will continue to consider this tool for future analyses.

Each of the specific edits that you listed below were corrected in the text. Thank you!

  1. Line 32. BVDV should be spelled-out.
  2. Line 51 and others. In vitro and in vivo should be in Italic.
  3. Line 108. MspI not Msp1.
  4. Lines 120-121. This sentence should be revised.
  5. Line 156. nL not nanoliters.
  6. Line 187. Student's should start with a capital letter.
  7. Lines 201-202. Delete differentially methylated regions ().
  8. Line 203 and others. Phospholipase C not phospholipase c.
  9. Line 289. Th not T helper.
  10. Line 402 and others. Rho should start with a capital letter.
  11. Lines 454-455. Delete myoclonus epilepsy of Unverricht-Lundborg type ().
  12. Lines 455-456. Delete amyotrophic lateral sclerosis ().
  13. Line 583. Influenza virus infection.
  14. Line 637. IFNs not interferons.
  15. Line 649. Zika virus (ZIKV).
  16. Lines 650-652. CMV not cytomegalovirus.
  17. Line 668. ZIKV.